# Risk Factor Analysis and a Predictive Model of Postoperative Depressive Symptoms in Elderly Patients Undergoing Video-Assisted Thoracoscopic Surgery

**DOI:** 10.3390/brainsci13040646

**Published:** 2023-04-11

**Authors:** Dinghao Xue, Xu Guo, Yanxiang Li, Zhuoqi Sheng, Long Wang, Luyu Liu, Jiangbei Cao, Yanhong Liu, Jingsheng Lou, Hao Li, Xinyu Hao, Zhikang Zhou, Qiang Fu

**Affiliations:** 1Medical School of Chinese PLA, Beijing 100853, China; 2Department of Anesthesiology, The First Medical Center, Chinese PLA General Hospital, Beijing 100853, China; 3Department of Pain Medicine, The First Medical Center, Chinese PLA General Hospital, Beijing 100853, China

**Keywords:** postoperative depressive symptoms, aged patients, video-assisted thoracoscopic surgery, risk factors, predictive model

## Abstract

Among the elderly, depression is one of the most common mental disorders, which seriously affects their physical and mental health and quality of life, and their suicide rate is particularly high. Depression in the elderly is strongly associated with surgery. In this study, we aimed to explore the risk factors and establish a predictive model of depressive symptoms 1 month after video-assisted thoracoscopic surgery (VATS) in elderly patients. The study participants included 272 elderly patients (age > 65 years) undergoing VATS from April 2020 to May 2021 at 1 of 18 medical centers in China. The patients were divided into a depression group and a nondepression group according to the Chinese version of the nine-item Patient Health Questionnaire (PHQ-9). The patients’ pre- and postoperative characteristics and questionnaires were collected and compared. Then, binary logistic regression was used to determine the risk factors that affect postoperative depressive symptoms, and the predictive model was constructed. The prediction efficiency of the model was evaluated by drawing the receiver operating characteristic curve (ROC), and the area under the curve (AUC) was calculated to evaluate the value of the predictive model. Among all of the included patients, 16.54% (45/272) suffered from depressive symptoms after VATS. The results of the univariate analysis showed that body mass index (BMI), chronic pain, leukocyte count, fibrinogen levels, prothrombin time, ASA physical status, infusion volume, anxiety, sleep quality, and postoperative pain were related to postoperative depressive symptoms (all *p* < 0.05). The results of multivariate logistic regression analysis showed that a high fibrinogen level (OR = 2.42), postoperative anxiety (OR = 12.05), poor sleep quality (OR = 0.61), and pain (OR = 2.85) were risk factors of postoperative depressive symptoms. A predictive model was constructed according to the regression coefficient of each variable, the ROC curve was drawn, and the AUC value was calculated to be 0.889. The prediction model may help medical personnel identify older patients at risk of developing depressive disorders associated with VATS and may be useful for clinical purposes.

## 1. Introduction

With the gradual aging of society, the mental health of the elderly has increasingly attracted attention. Among the elderly, depression is one of the most common mental disorders that seriously affect their physical and mental health and quality of life [1]. Risk factors for depression, especially in the elderly, are associated with inflammation, surgery, gender, social support, lifestyle, etc. [2,3,4,5,6]. Elderly patients are more susceptible to developing multiple comorbidities, such as cancer, chronic obstructive pulmonary disease, and coronary heart disease. Consequently, they are more likely to undergo complex surgeries, which may result in physiological stress and pain.

Surgery triggers negative psychological emotions, intense psychological stress reactions, and severe physical trauma, especially in elderly patients [7]. Additionally, elderly people may face various psychological factors such as retirement and the loss of social status, which may lead to a lowering of their self-esteem, increasing psychological pressure, and a triggering of depressive symptoms. Anesthetics and analgesics can affect a patient’s mood and mental state, increasing the risk of postoperative depression. Elderly patients may experience isolation and loneliness after surgery, especially if they are unable to immediately return to their normal daily activities. This can lead to feelings of helplessness, a loss of control, and anxiety, all of which can directly contribute to postoperative depression.

Video-assisted thoracoscopic surgery (VATS) is a minimally invasive surgical technique: a combination of video technology and endoscopic technology [8]. VATS is commonly used for diagnostic and therapeutic purposes, including lung biopsy, pleural biopsy, and lobectomy. VATS can reduce the trauma and complications of thoracic surgery and improve patients’ quality of life after surgery [9]. VATS is a safe and effective surgical option for many patients with chest-related conditions. Currently, minimally invasive technology is a novel trend in clinical diagnosis and treatment. The risk factors for depressive symptoms after VATS may differ from those of traditional thoracotomy. 

Researchers have focused on the difference in postoperative depressive symptoms between patients undergoing thoracotomy and VATS [10,11,12]. Patients who underwent VATS showed a significant reduction in postoperative pain and an improvement in their quality of life, while those who underwent anterolateral thoracotomy did not. This may be due to the less invasive and painful nature of VATS, which reduces the physical and psychological stress on patients and has an antidepressant effect [13]. However, not all studies support the relationship between VATS and antidepressant effects. Another study suggested that VATS was not significantly different from traditional open surgery and did not have a specific impact on patients’ depressive symptoms [14]. Most research on the relationship between VATS and depression to date have been cross-sectional, which precludes causal inferences due to the absence of temporal precedence [15] to determine the risk factors of postoperative depressive symptoms. 

To the best of our knowledge, a series of studies have been conducted on the risk factors for depressive symptoms after open surgery, whereas no study has been undertaken on postoperative depressive symptoms in elderly patients after VATS. Therefore, we conducted a retrospective multicenter study to evaluate these risk factors and constructed a risk prediction model for depressive symptoms in patients older than 65 years 1 month after VATS. 

## 2. Materials and Methods

### 2.1. Study Design

All questionnaires were completed by an experienced anesthesiologist or anesthesia nurse who was blinded to the protocol during the patient visit. On the day before surgery, specialized medical staff conducted face-to-face interviews with patients and completed the Chinese version of the Mini-Mental State Examination (MMSE) (Appendix A), the Chinese version of the nine-item Patient Health Questionnaire (PHQ-9) (Appendix A), and the Chinese version of the seven-item Generalized Anxiety Disorder Questionnaires (GAD-7) (Appendix A). We conducted telephone follow-ups with the patients and completed the PHQ-9, GAD-7, numeric rating scale (NRS), and quality of sleep questionnaires 1 month after surgery.

This prediction model study is reported in accordance with the Transparent Reporting of a multivariable prediction model for Individual Prognosis Or Diagnosis (TRIPOD) checklist [16].

### 2.2. Study Participants

This is a multicenter, retrospective study based on the Perioperative Database of Chinese Elderly Patients (PDCEP). The PDCEP is a database of elderly patients in China during the perioperative period. This project was led by the National Clinical Research Center for Geriatric Diseases (Chinese PLA General Hospital) and collected data from 18 centers in 11 regions of China. The PDCEP aimed to facilitate the investigation of perioperative complications in elderly patients. 

The study population included elderly patients undergoing VATS from April 2020 to May 2021 at 1 of 18 medical centers in this database. The inclusion criteria were as follows: (1) age ≥ 65 years; (2) general anesthesia with tracheal intubation; (3) patients without cognitive dysfunction. The cognitive function of the elderly was measured with the MMSE. Participants whose MMSE scores are lower than the corresponding lowest scores for their education level are considered to have cognitive dysfunction: scores below 18 for those without formal education, scores below 21 for those with 6 or fewer years of education, and scores below 25 for those with more than 6 years of education. Patients were excluded from the analysis if they (1) exhibited preoperative depressive symptoms (preoperative PHQ-9 score ≥ 5) or anxious symptoms (preoperative GAD-7 score ≥ 5), (2) were treated with or converted to open thoracotomy, and (3) were treated with robot-assisted surgery. We used PHQ-9 to screen the depressive symptoms of patients 1 month after surgery. A total of 272 patients were divided into a depression group (*n* = 45) and a nondepression group (*n* = 227) according to PHQ-9 (Figure 1). The PHQ-9 questionnaire consists of nine items, each with four response options: “not at all”, “several days”, “more than half of the days”, and “almost every day”, corresponding to scores of 0, 1, 2, and 3, respectively. The total score of the questionnaire is 27. A score of 0 to 4 suggests no depressive symptoms; scores ≥ 5 indicate the presence of depression symptoms. A score of 5 to 9 suggests mild depressive symptoms, one of 10 to 14 suggests moderate depressive symptoms, and one of 15 to 27 suggests severe depressive symptoms.

### 2.3. Ethics Statement

This study, which involved human participants, was reviewed and approved by the Ethics Committee Board of the First Medical Center of Chinese PLA General Hospital (No. S2019-311-02).

### 2.4. Anesthesia and Analgesia Protocols

During anesthesia induction, a sequence of intravenous injections was administered, including midazolam at a dose of 0.05 mg/kg, etomidate at a dose of 0.3 mg/kg or propofol at a dose between 2.0 and 2.5 mg/kg, sufentanil at a dose between 0.2 and 0.4 g/kg, and rocuronium at a dose of 0.6 to 0.9 mg/kg. Oxygen was provided via a face mask during the induction period, followed by mechanical ventilation after successful tracheal intubation. Intravenous–inhalational combined anesthesia was used to maintain anesthesia during the surgery, with 1–3% sevoflurane inhalation and the maintenance of the end-tidal sevoflurane concentration at 0.7–1.5 MAC. Propofol was continuously infused intravenously at a dose of 2–4 mg/kg/h, remifentanil was infused at a dose of 0.1–0.2 μg/kg/h, and intermittent intravenous injections of rocuronium and sufentanil were administered to maintain anesthesia. 

Patient-controlled intravenous analgesia (PCIA) was used, with a formula consisting of 2.0 μg/kg sufentanil, 24 mg ondansetron, and 0.9% sodium chloride injection, with a dilution up to 100 mL. The background dose was set to 1.5 mL/h, and the single dose was set to 1 mL, with a lockout time of 15 min.

### 2.5. Data Collection 

Basic and clinical demographic information was collected, including age, sex, body mass index (BMI), education level, alcohol drinking, smoking, comorbidities (hypertension, diabetes, chronic obstructive pulmonary disease (COPD), coronary heart disease, and cerebrovascular disease), chronic pain, exercise tolerance, and preoperative fasting time. The collected laboratory test results included serum albumin, hemoglobin, erythrocyte count, leukocyte count, serum glucose, blood urea nitrogen, blood potassium, blood sodium, blood chlorine, fibrinogen, prothrombin time, aspartate aminotransferase (AST), and alanine aminotransferase (ALT). The intraoperative data included the American Society of Anesthesiologists (ASA) physical status, temperature monitoring, surgery duration, blood loss, urine output, infusion volume, and postoperative analgesia. The data obtained 1 month after VATS that we collected included anxiety, sleep quality, and postoperative pain. 

The GAD-7 was used to screen anxiety symptoms. The GAD-7 questionnaire comprises seven items, each of which has four response options: “not at all”, “several days”, “more than half of the days”, and “almost every day”, corresponding to scores of 0, 1, 2, and 3, respectively. The total score of the questionnaire is 21. A score of 0 to 4 suggests no anxious symptoms, and scores ≥ 5 indicate the presence of anxious symptoms. A score of 5 to 9 suggests mild anxious symptoms, one of 10 to 14 suggests moderate anxiety, and one of 15 to 21 suggests severe anxious symptoms.

The quality of sleep was measured by a scale by which patients chose a number from 0 to 10 that they felt best described their quality of sleep the previous night. The quality of sleep was primarily evaluated based on three aspects: difficulty falling asleep, shallow sleep, and early awakening.

To determine whether a patient had chronic pain, we inquired about the presence of pain lasting for 3 months or longer.

Postoperative pain measurement was scored using the NRS. Zero usually represents “no pain at all”, whereas the upper limit represents “the worst pain imaginable”. We employed the NRS to assess pain in patients under both rest and activity conditions, with a score of 0 indicating the absence of pain in either state. 

### 2.6. Statistical Analysis

All statistical analyses and data management were performed in SPSS version 26 (IBM Corporation, Armonk, NY, USA). The jointly normally distributed data are reported as the mean ± standard deviation (SD). Continuous non-normal data are reported as the median and interquartile range (IQR). Categorical data are expressed as the frequency and proportion. The *t*-test, the Mann–Whitney *U* nonparametric test, the chi-square test, and the continuity adjusted chi-square test were used to compare the differences between patients with and without postoperative depressive symptoms. The confounding variables with significant differences were then included in the binary logistic regression model to determine the risk factors for depressive symptoms and to establish a risk model. The receiver operating characteristic (ROC) curve was obtained by the prediction model based on logistic regression, and the area under the curve (AUC) and cut-off value were calculated; *p* < 0.05 indicated a statistically significant difference, and *p* values were two-sided.

## 3. Results

### 3.1. Preoperative Basic and Clinical Demographics of Patients 

A total of 272 patients met the study inclusion criteria: 45 patients reported postoperative depressive symptoms among the 272 patients, and 227 reported no depression. The group reporting postoperative depressive symptoms had a higher BMI and a higher proportion of preoperative chronic pain (all *p* < 0.05). Additionally, the elderly patients with postoperative depressive symptoms had a higher leukocyte count, fibrinogen level, and prothrombin time than those without postoperative depressive symptoms (all *p* < 0.05). See Table 1 for details.

### 3.2. Intraoperative and Postoperative Characteristics of the Two Groups 

Patients with postoperative depressive symptoms had a significantly lower proportion of ASA physical status > II (*p* = 0.03). Regarding intraoperative fluid management, the patients with postoperative depressive symptoms had a larger infusion volume than the patients without postoperative depressive symptoms (*p* = 0.03). The patients with postoperative depressive symptoms had a higher proportion of anxiety and pain and poorer sleep quality 1 month after surgery (all *p* < 0.05), as shown in Table 2.

### 3.3. Multivariate Logistic Regression Analysis

A binary logistic regression model that included the significant variables listed in Table 1 and Table 2 was used to identify independent risk factors and establish a risk model for postoperative depressive symptoms. The results indicated that a high fibrinogen level (OR = 2.42), postoperative anxiety (OR = 12.05), poor sleep quality (OR = 0.61), and pain (OR = 2.85) were significantly correlated with a higher risk of postoperative depressive symptoms; see Table 3.

### 3.4. Evaluation of the Prediction Model

The ROC curve (Figure 2) was used to verify the established risk prediction model. The results showed that the AUC of the logistic regression model was 0.889 (95% CI: 0.844–0.934, *p* < 0.001). The cut-off value was 0.661, the sensitivity was 95.24%, and the specificity was 90.81%. The Hosmer–Lemeshow test results showed that the Hosmer–Lemeshow *χ*^2^ = 15.31 (*p* = 0.053). We found that the fitting degree of the model was favorable. 

## 4. Discussion

We explored the PDCEP to determine the independent risk factors of and develop a prediction model for postoperative depressive symptoms of older persons after VATS. We first conducted univariate analysis of depressive symptoms in elderly patients. These variables were then incorporated into binary logistic regression models to establish a risk model and identify independent risk factors for depressive symptoms. The risk factors for depressive symptoms determined 1 month after VATS in the aged were a preoperative high level of fibrinogen, postoperative anxiety, poor sleep quality, and pain. 

Postoperative depression is pervasive and is one of the major causes of postoperative mental health issues [2,17,18]. Therefore, the mental trauma of surgery cannot be ignored, especially in elderly patients. The high incidence of affective disorders in elderly patients undergoing VATS raised questions regarding the etiology of depression in this population. Although the etiology of depression is still unclear, it is likely due to insomnia, comorbid disorders, pain, inflammation, cognitive function, sex, and social support [19,20,21,22,23,24]. Consequently, we need to study the risk factors of the disease.

A link was found between elevated fibrinogen levels and depressive symptoms [25,26]. Similarly, we found that a preoperative higher fibrinogen level increased the risk of depressive symptoms 1 month after surgery in the elderly patients. Matthews et al. [27] found that an elevated fibrinogen level was related to depressive symptoms among women approaching menopause. Qiu et al. [28] also found that elevated plasma fibrinogen levels at admission increased the risk of developing depression 3 months after stroke in men. The two studies indicated that a high level of fibrinogen may be, in part, associated with depressive symptoms through hypercoagulability. Research findings based on bioinformatics and proteomics suggested that the fibrinogen alpha chain (FGA) may be a biomarker for the diagnosis of depression and a biological target for depression [29,30]. In addition, researchers are increasingly focusing on the role fibrinogen plays in inflammation and depression. Fibrinogen can increase proinflammatory cytokine levels by stimulating the synthesis of proinflammatory cytokines, which can disturb tryptophan metabolism [31,32,33,34]. Then, there would be a decrease in serotonin, which plays a crucial role in the development of depression, because tryptophan is a precursor of serotonin [35].

In our study, anxiety was a risk factor for postoperative depressive symptoms; this finding is similar to those of previous studies in elderly patients. A multiwave prospective study showed that anxious arousal symptoms led to depressive symptoms in adolescents [36]. John et al. [37] reported that prior anxiety increased the risk of depression in both sexes. Estimates of the prevalence of anxiety in older adults were 15–52%, and that of the prevalence of comorbid anxiety disorders in older adults with depression was 38.6% [38,39]. The rate of suicide is higher in elderly adults with anxiety superimposed upon depression than in those with depression only [20,40].

Sleep disturbance is a risk factor for depression disorders among the aged [41]. Moreover, sleep disturbance is fairly common among the elderly [42]. Insomnia, excessive daytime sleepiness, and sleep medication were independently associated with the risk of depression in the elderly population [43]. Surgery is known to be a risk factor for poor sleep, and a host of patients suffer from perioperative sleep disturbance [44,45]. Pain and anxiety were positively associated with preoperative and postoperative sleep disturbance [46,47]. We investigated the relationship between the participants’ subjective sleep quality and found that poor sleep quality was an independent risk factor of postoperative depressive symptoms. A long-term follow-up study is needed, and possible associations with other factors concerning sleep should be investigated to improve our understanding of these patients.

Pain is defined as an unpleasant sensory and emotional experience associated with actual or potential tissue damage or is described in terms of such damage [48]. A pooled analysis of two national aging cohort studies showed that a bidirectional association exists between pain and depressive symptoms. People with pain were at a higher risk of suffering from depressive disorders and vice versa [49]. Our study showed that postoperative pain was an independent risk factor for depressive symptoms. Furthermore, we found that the most painful sites were the surgical incision or the location of the drainage tube. Silvia et al. [50] reported that the incidence of chronic postsurgical pain (CPSP) caused by VATS was as high as 35%. This indicated that postoperative pain has a high probability of developing into CPSP. Peter et al. [51] reported that pain precedes the onset of depression, and chronic pain in the elderly places them at risk of depression. Therefore, specific strategies for preventing pain are needed to prevent depression. 

We developed a predictive model for identifying the risk factors of postoperative depressive symptoms in elderly VATS patients. The predictive model demonstrated a strong predictive ability; the AUC of the logistic regression model was 0.889, the cut-off value was 0.661, the sensitivity was 95.24%, and the specificity was 90.81%. The goodness of fit of the predictive model was assessed through the Hosmer–Lemeshow test, revealing a Hosmer–Lemeshow *χ*^2^ = 15.31 (*p* = 0.053). This suggested that the predictive model fit the observed data well and highlighted its high accuracy in predicting the risk of postoperative depressive symptoms. The application of this predictive model for the early identification of high-risk individuals for postoperative depressive symptoms can assist healthcare professionals in selecting more appropriate treatment modalities and improving patient prognosis.

### Strengths and Limitations

This study makes several contributions to the current literature. First, this is one of the earliest clinical studies published on the PDCEP, which will serve as a crucial reference for future research on the database. Second, to the best of our knowledge, this is the first predictive model specifically designed to predict postoperative depressive symptoms in elderly VATS patients. The identification of relevant risk factors can be highly beneficial for guiding anesthesiologists in perioperative medication management, which can be highly beneficial for patients.

However, this study is not without some limitations. First, this prospective study included both preoperative and postoperative predictors in the final multivariate regression equation. Although a rigorous design was adopted, possible bias could have interfered with the research results and conclusions. Large prospective multicenter studies are essential for confirming the model and results. Second, the IQRs of fibrinogen were in the 0.8 range and the normal range. This may provide fewer indications for the individual patient. However, this demonstrated the differential trend of this indicator between the two groups. The inclusion of more participants in a future study can further elucidate the role of fibrinogen. Third, for the complex phenomenon of sleep, the 0–10 sleep score may be somewhat simple. In an ideal scenario, patients would assess their sleep quality through the use of wearable sleep monitoring devices, and we are prepared to undertake such studies. Finally, the analyses did not statistically adjust for the heterogeneity of effects due to the multiple centers. We would conduct multilevel logistic regression and related analyses in future multicenter replication and extension studies with more extensive and well-powered sample sizes [52].

## 5. Conclusions

In our study, we explored risk factors for depressive symptoms 1 month after VATS in elderly patients from a multicenter database and established a predictive model. Reducing preoperative fibrinogen levels and improving postoperative anxiety, sleep quality, and pain are of considerable benefit to patients. Prediction enables proactive intervention. The proposed predictive model may help medical personnel identify older patients at risk of developing depressive disorders associated with VATS and may be useful for clinical purposes.

## Figures and Tables

**Figure 1 brainsci-13-00646-f001:**
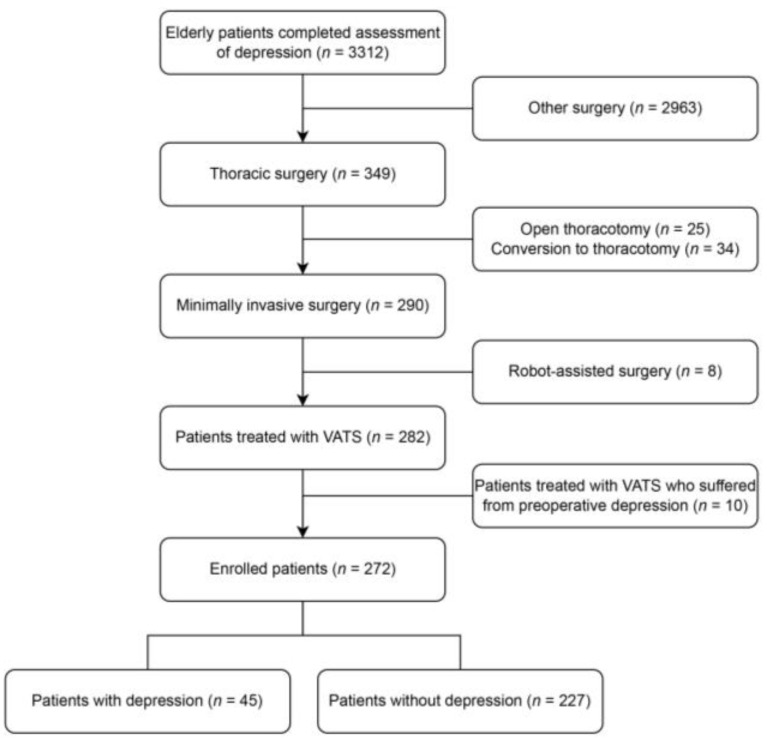
Details on study recruitment.

**Figure 2 brainsci-13-00646-f002:**
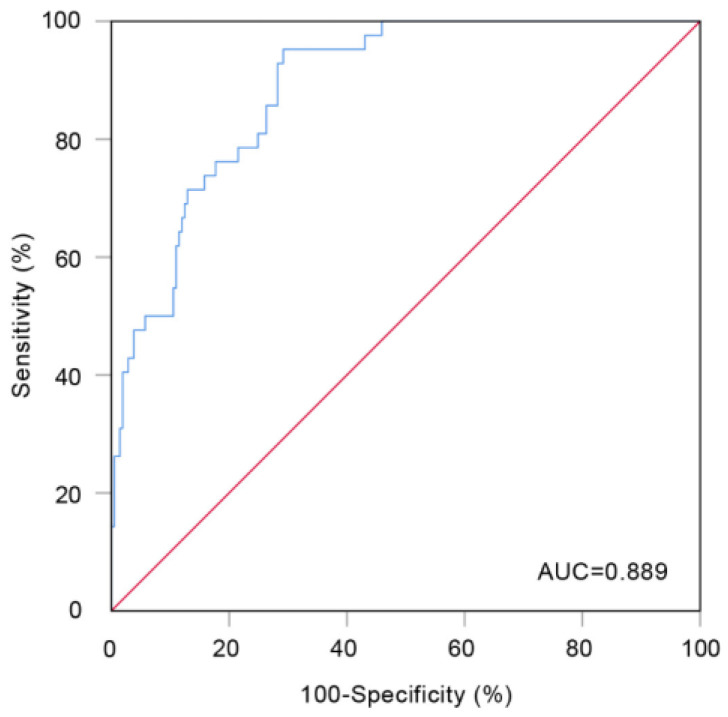
Receiver operating characteristic curve (ROC) obtained by the risk model based on logistic regression.

**Table 1 brainsci-13-00646-t001:** Preoperative basic and clinical demographics of patients with and without depressive symptoms.

Variable	Depression Group (*n* = 45)	Nondepression Group(*n* = 227)	*χ*^2^/*Z*/*t* Value	*p* Value
Age (years) (IQR)	69 (67, 72)	69 (67, 72)	−0.41	0.68
Sex (male) (*n*, %)	15 (33.33)	28 (22.05)	2.26	0.16
BMI (kg/m^2^) (IQR)	25.39 (23.33, 26.67)	24.00 (22.31, 26.04)	−2.17	0.03
Education level (years > 8) (*n*, %)	19 (42.22)	118 (51.98)	1.43	0.26
Alcohol drinker (*n*, %)	12 (26.67)	75 (33.04)	0.70	0.49
Smoking (*n*, %)	11 (24.44)	81 (35.68)	2.12	0.17
Comorbidities				
Hypertension (*n*, %)	20 (44.44)	99 (43.61)	0.01	>0.99
Diabetes (*n*, %)	9 (20.00)	41 (18.06)	0.09	0.83
COPD (*n*, %)	3 (6.67)	17 (7.49)	0.00	>0.99
Coronary heart disease (*n*, %)	10 (22.22)	43 (18.94)	0.26	0.68
Cerebrovascular disease (*n*, %)	2 (4.44)	12 (5.29)	0.00	>0.99
Chronic pain (*n*, %)	6 (13.33)	7 (3.08)	6.56	0.01
Exercise tolerance (METs < 5) (*n*, %)	11 (24.44)	37 (16.30)	1.71	0.20
Preoperative fasting time (h) (IQR)	12 (8, 12)	10 (8, 13)	−0.41	0.68
Laboratory testing				
Serum albumin (g/L) (SD)	43.07 (3.12)	41.86 (3.82)	2.00	0.05
Hemoglobin (g/L) (SD)	134.51 (15.23)	135.21 (14.31)	0.30	0.77
Erythrocyte count (×10^12^/L) (SD)	4.19 (1.02)	4.31 (0.75)	0.87	0.39
Leukocyte count (×10^9^/L) (IQR)	6.32 (5.15, 7.32)	5.52 (4.58, 6.44)	−2.52	0.01
Serum glucose (mmol/L) (IQR)	5.33 (4.89, 5.88)	5.19 (4.76, 5.86)	−1.13	0.26
Blood urea nitrogen (mmol/L) (IQR))	5.48 (4.44, 6.60)	4.49 (4.56, 6.61)	−0.16	0.87
Blood potassium (mmol/L) (SD)	4.08 (0.34)	4.08 (0.35)	0.00	>0.99
Blood sodium (mmol/L) (IQR)	141.65 (140.01, 142.88)	142.00 (140.70, 143.38)	−1.40	0.16
Blood chlorine (mmol/L) (SD)	103.90 (2.98)	104.45 (2.94)	1.14	0.25
Fibrinogen (g/L) (IQR)	3.32 (2.95, 3.79)	2.92 (2.50, 3.31)	−3.93	<0.001
Prothrombin time (s) (IQR)	12.75 (11.65, 13.40)	12.10 (11.00, 13.10)	−2.17	0.03
AST (U/L) (IQR)	18.10 (15.63, 21.88)	16.30 (14.30, 20.40)	−1.87	0.06
ALT (U/L) (IQR)	15.65 (10.80, 21.78)	14.90 (11.50, 20.20)	−0.49	0.63

BMI: body mass index; COPD: chronic obstructive pulmonary disease; METs: metabolic equivalents; AST: aspartate aminotransferase; ALT: alanine aminotransferase; IQR: interquartile range; SD: standard deviation.

**Table 2 brainsci-13-00646-t002:** Intraoperative and postoperative characteristics of patients with and without depressive symptoms.

Variables	Depression Group (*n* = 45)	Non-Depression Group (*n* = 227)	*χ*^2^/*Z*/*t* Value	*p* Value
ASA physical status (>II), (*n*, %)	3 (6.67)	47 (20.70)	4.93	0.03
Temperature monitor (*n*, %)	42 (93.33)	208 (91.63)	0.01	0.93
Surgery duration (min) (IQR)	137 (85, 137)	125 (91, 155)	−0.35	0.73
Blood loss (mL) (IQR)	50 (20, 60)	50 (20, 50)	−1.19	0.23
Urine output (mL) (IQR)	260 (100, 500)	300 (100, 500)	−0.17	0.87
Infusion volume (mL) (IQR)	1600 (1100, 1604)	1100 (1100, 1600)	−2.21	0.03
Postoperative analgesia (*n*, %)	37 (82.22)	178 (78.41)	0.33	0.69
Anxiety (*n*, %)	12 (26.67)	9 (3.96)	24.07	<0.001
Sleep quality (IQR)	6.00 (5.00, 7.50)	8.00 (7.00, 9.00)	−5.53	<0.001
Pain (*n*, %)	31 (68.86)	87 (38.33)	14.28	<0.001

ASA: American Society of Anesthesiologists; IQR: interquartile range.

**Table 3 brainsci-13-00646-t003:** Multivariate analysis of risk factors for depressive symptoms in elderly patients undergoing VATS.

Factor	β	SE	Wald	*p* Value	OR Value	95% CI
BMI	0.10	0.07	1.94	0.16	1.10	0.96~1.26
Chronic pain	1.29	0.83	2.40	0.12	3.62	0.71~18.42
Leukocyte count	0.08	0.15	0.30	0.59	1.08	0.81~1.45
Fibrinogen	0.88	0.31	7.88	0.01	2.42	1.30~4.47
Prothrombin time	0.01	0.12	0.01	0.91	1.01	0.80~1.28
ASA physical status	−0.90	0.77	1.37	0.24	0.41	0.09~1.84
Infusion volume	0.00	0.00	1.26	0.26	1.00	1.00~1.00
Anxiety	2.49	0.65	14.74	<0.001	12.05	3.38~42.95
Sleep quality	−0.49	0.12	15.99	<0.001	0.61	0.48~0.78
Pain	1.05	0.44	5.56	0.02	2.85	1.19~6.81

BMI: body mass index; ASA: American Society of Anesthesiologists; CI: confidence interval; OR: odds ratio; SE: standard error.

## Data Availability

Not applicable.

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
