# Peer review of "Risk Factor Analysis and a Predictive Model of Postoperative Depressive Symptoms in Elderly Patients Undergoing Video-Assisted Thoracoscopic Surgery"

_brainsci, 2023, doi:10.3390/brainsci13040646_

Round 1
Reviewer 2 Report
This is a very interesting paper investigating risk factors and predictive factors of posoperative depression in elderly patients undergoing video-assisted thoracoscopic surgery. The paper is well-written and of interest for the journal; however, several minor changes are recommended before considering it for publication.
Abstract.
1- Depression is (of course) affecting mental health. This is stated in the abstract. Please, provide a sentence more focused on the clinical outcomes (functionality, etc).
2-How were the patients recruited? I recommend to describe the study design. This is important for the readers to interpret the results.
3- The conclusion of the abstract should be more general. I do not recommend to present a summary of the results.
Materials and methods.
1- Before describing "This prediction model study is reported in according with the TRIPOD checklist", I recommend to describe the study design.
2- The Ethics statement can be described after the section patients. I recommend to rename "Patients" by "Study Participants".
Results
1- The main characteristics of the sample, and the main results are weel-described.
Discussion
1- The last part focusing on the limitations of the study merits a subsection called 4.1. Strenghts and limitations.
Conclusions
1- The conclusions section can not be started by "sum up". The conclusions are not a summary of the results. Please, revise this section. What about expanding with future perspectives?
Reviewer 3 Report
The study aimed to evaluate risk factors for postoperative depression in elderly patients as well as the development of the risk prediction model. The study presents high scientific value, but some minor aspects should be addressed:
- I think more background information about depression (especially in elderly patients) and VATS should be presented in the introduction.
- Is there any protocol for that study? Could the authors provide a link to it? Was it uploaded somewhere? If not, that information could be presented in a supplementary file.
- The supplementary file could also contain MMSE and PHQ-9 questionaries.
- The inclusion and exclusion criteria should be more clearly presented for better visibility because this is the most crucial part of this experiment (e.g. bullet points).
- The statistical significance in Tables 1, 2 and 3 could be bolded (more visible).
- Model effectiveness/efficacy should also be discussed in the discussion section of the manuscript.
Reviewer 4 Report
The paper submitted for review concerns the search for risk factors for the development of depression in patients undergoing VATS. The manuscript has the correct structure, introduction is short but rich in content, it covers the topic. The authors clearly presented the main aim of the study. Some points in material and methods require some additions (information below). Statistical analysis was described in great detail. The approval number of the bioethics committee for the study is also displayed. The results are consistent, presented in a clear way using tables. The discussion is in the correct form, it contains numerous citations regarding similar studies. The conclusions drawn coincide with the obtained results.
Major comments
1. One of the exclusion criteria is preoperative depression. Could authors provide more information about this as it seems important. Were patients never suffered from depression or are they just currently unaffected? How was it checked? What about anxiety disorders?
2. To my knowledge, the PHQ-9 is an initial screening tool for depressive symptoms that requires confirmation by the clinician. Diagnosis cannot by made on the basis of this questionnaire. Were the diagnosis confirmed in any other way? Such information should be added to the manuscript. It would also be worth adding the questionnaire itself (to supplementary materials) and how it was assessed.
Minor comments
1. "Although previous studies have been sufficient, risk factors for depression in elderly surgery patients remain underexplored". This sentence should be changed/clarify.
2. There are some editorial/typing errors, for example in line 63 (no space), line 16 (two spaces). Please check the whole manuscript once again.
3. Are there any criteria made by authors to select the laboratory parameters which will be included in the statistical analysis?
4. "The prevalence of postoperative depression is higher than before VATS." This sentence from discussion must be clarified. What did the authors mean exactly?
Round 2
Reviewer 1 Report
I have looked through the revised mansuscript and the authors replies to my comments (no other referees involved with this paper? In case, what do he/she say?)
and I do not think they can change the major shortcomings in the design and results collection with this manuscript.
Author Response
Dear Reviewer,
Thank you for taking the time to review our manuscript. We appreciate your thorough evaluation and insightful comments.
We have carefully considered your feedback, particularly your concern about the major shortcomings in the design and results collection of our study. While we agree that these are important issues that could impact the validity of our results, unfortunately, we are unable to make further significant changes at this stage due to limitations in our resources and time constraints.
However, we have taken your feedback seriously and will take your suggestions into account for future research. We value your expertise and appreciate the time you have spent assessing our manuscript.
Thank you again for your valuable feedback.
Sincerely,
Qiang Fu and co-authors
Reviewer 3 Report
I am satisfied with the correction made
Author Response
Dear Reviewer,
We want to express our sincere thanks for your careful and thoughtful review of our manuscript. We greatly appreciate the time and effort you have dedicated to providing valuable feedback.
We are pleased to inform you that we have addressed the concerns raised in your previous review and have made significant corrections and improvements to the manuscript. We are grateful for your scrutiny and feedback, which have allowed us to strengthen our work and ensure its accuracy and reliability.
We are pleased to note that you are satisfied with the corrections made in the manuscript. We are glad that our revisions have been successful in addressing your concerns and we hope that the manuscript now meets the high standards of your esteemed journal.
Thank you again for your insightful feedback and for your valuable support of our research.
Sincerely,
Qiang Fu and co-authors